# Organic Material for Clean Production in the Batik Industry: A Case Study of Natural Batik Semarang, Indonesia

**Nana Kariada Tri Martuti** [1,*] **, Isti Hidayah** [2] **, Margunani Margunani** [3] **and Radhitya Bayu Alafima** [4]

[1] Biology Department, Faculty of Mathematic and Natural Sciences, Universitas Negeri Semarang, 1st Floor, D6 Building Kampus Sekaran Gunungpati, Semarang 50229, Indonesia

[2] Mathematics Department, Faculty of Mathematic and Natural Sciences, Universitas Negeri Semarang, 1st Floor, D5 Building Kampus Sekaran Gunungpati, Semarang 50229, Indonesia; isti.hidayah@mail.unnes.ac.id

[3] Economic Department, Faculty of Economi, Universitas Negeri Semarang, C6 Building, Kampus Sekaran. Gunungpati, Semarang 50229, Indonesia; margunani@mail.unnes.ac.id

[4] Department of Industrial Technology of Agriculture, Faculty of Agricultural Technology, Institut Pertanian Bogor, Bogor 16680, Indonesia; radbaya123@gmail.com

* Correspondence: nanakariada@mail.unnes.ac.id

**Abstract:** Batik has become more desirable in the current fashion mode within the global market, but the environmental damage induced by this fabric's synthetic dye practices is a matter of concern. This study aimed to discuss the application of organic materials as natural dyes in the clean production of textiles to maintain the environment. The research was a case study from the community services program in Kampung Malon, Gunungpati, Semarang City, Indonesia, focused on the batik home industry of the Zie Batik fabric. Furthermore, natural pigments from various plant organs (stem, leaves, wood, bark, and fruit) of diverse species, including *Caesalpinia sappan*, *Ceriops candolleana*, *Maclura cochinchinensis*, *Indigofera tinctorial*, *I. arrecta*, *Rhizopora* spp., *Strobilantes cusia*, and *Terminalia bellirica* were used for this type of material. These pigments are more biodegradable, relatively safe, and easily obtained with zero liquid waste compared to the synthetic variants. The leftover wastewater from the coloring stages was further utilized for other processes. Subsequently, the remaining organic waste from the whole procedure was employed as compost and/or timber for batik production, although a large amount of the wastewater containing sodium carbonate ($Na_2CO_3$), alum ($KAl(SO4)_2 \cdot 12H_2O$), and fixatives ($Ca(OH)_2$ and $FeSO_4$) were discharged into the environment during the process of mordanting and fixating, with the requirement of additional treatment.

**Keywords:** batik; clean production; natural; organic materials; Semarang City

## 1. Introduction

The Indonesian traditional batik is recognized as a masterpiece within the oral and intangible heritage of humanity by UNESCO [1,2]. This material's uniqueness is seen from the pattern and variety of motifs illustrating nature, diversity of fauna and flora, folktales, as well as weapons [1–4]. The fabric global market was dominated by this material in 2017, with its export value reaching USD 58.46 million, where the main destinations included Japan, the United States, and European countries. Furthermore, issues of synthetic dye practices have generated a decline in the selling demand of this fabric in several destination countries. This is because the usage of manmade variants of synthetic dyes potentially generate serious problems for animal and human health [5], including cancer [6,7], as well as polluting

water sources [8] and disturbing organisms and ecosystem balance [9,10]. Furthermore, various studies have shown an extremely high content of dangerous heavy metals, including Cd, Cr [11], Cu, and Pb, around areas of the batik industry [12,13].

The poor awareness of batik makers on environmental sustainability is the reason synthetic dyes are massively used without the regulation of standardized waste management [14]. In some cases, including the synthetic batik industry in Pekalongan [8,9] and Surakarta [15], wastewater was produced with dangerous amounts of heavy metals above the environmentally permitted standard.

The momentum of the Indonesian batik export growth is simultaneously a challenge and an opportunity. Furthermore, global consumer demand for environmentally friendly (eco-friendly) products emerged as a response to the green lifestyle and environmental awareness movement. The use of these types of materials is a universal consumption trend, adopted as an effort to create a harmonious life between nature (green lifestyle) and the batik industry [16,17]. This sector has a great opportunity to manufacture eco-friendly products through the application of organic materials in batik dyes, which would be a cleaner production method.

Zie Batik is a small and medium-scale enterprise (SME) actively producing fabrics with natural dyes. This company is located in Gunungpati Subdistrict, Semarang City, with total revenue of IDR 30–40 million or USD 2100.00–2800.00/month, and is the iconic landmark of this region. Furthermore, production commenced by developing authentic batik motifs in 2006. The Zie Batik is also the first SME developing and introducing organic dyes in Semarang City, and the pioneer for standardized batik in Indonesia. Capturing the information from Zie Batik's production process is essential for disseminating proper eco-friendly batik business.

It cannot be denied that many have researched natural dyes in fabric, but their application in several places should be invented to gain a comprehensive understanding of batik dye based on the local potential. Then, even though small to medium-scale enterprises (SMEs) are well known as profit-oriented businesses, we explain this new perspective of the batik natural dye industry related to cleaner production in SMEs, especially for Semarang City as an industrial and metropolitan city. The process commences with organic pigment application and waste treatments. This form of dye prevents and integrates environmental management strategies without ignoring aspects of economic and cultural development as a sustainable development approach. The aim of this study was to comprehensively describe the clean production application of natural dyes produced from local value chain organic material as an alternative solution in this sector to maintain the environment. We also propose and depict how the SME in this study developed a batik industry with a sustainable development approach.

## 2. Results

### 2.1. The Batik Motif and Organic Material for Natural Dye

Zie Batik developed more than 10 authentic motifs or characters for their batik signature from Semarang City, with the famous one being the Javanese Puppet batik, which tells Ramayana's story. The city's iconic landmarks, portraying the historical buildings, including Tugu Muda, Lawang Sewu, Pagoda, and Warak Ngendog—the city mascot—was developed by Zie Batik to introduce natural, high-quality batik using clean production, widely used in the batik home industry in Semarang City (Figure 1). The growing demand for natural fabrics as well as the intense competition has led Zie Batik to develop other motifs illustrating fauna, flora, and the legend of the Javanese Puppet stories, as depicted in Figure 1B.

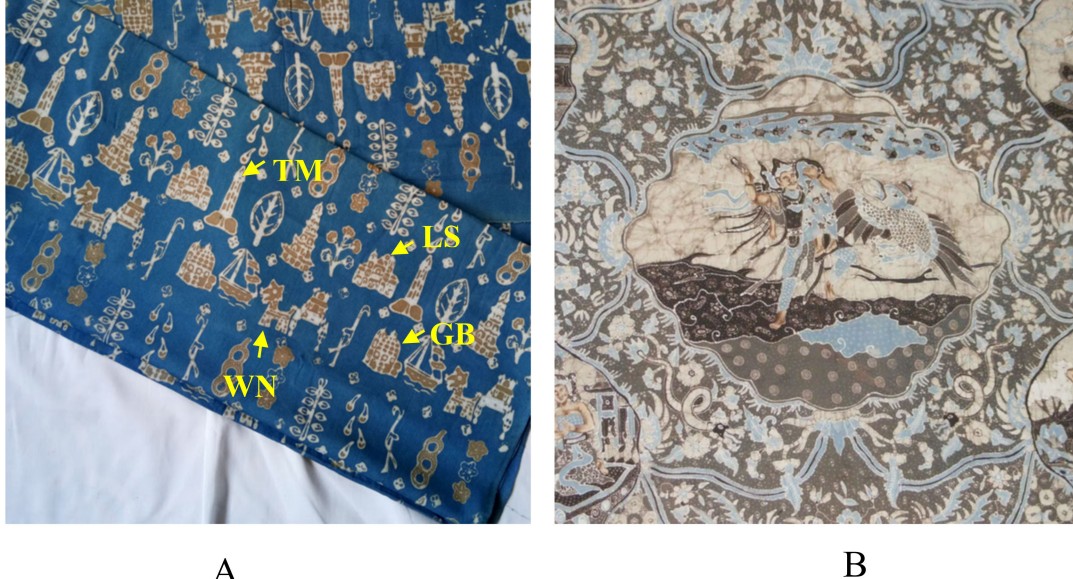

A　　　　　　　　　　　　　　　　　　　　　　　　　B

**Figure 1.** The most popular motifs of the Zie Batik products by themes. The iconic landmark of Semarang City or more popular with Semarangan motifs (**A**) and the legend of the Javanese Puppet stories (**B**). In the Semarangan motifs, the natural dye batik is fully decorated with the iconic buildings of Semarang City, such as Tugu Muda (TM), Lawang Sewu (LS), and Gereja Blendug (GB), as well as the city mascot Warak Ngendok (WN), a well-known mythological animal in Semarang City.

Furthermore, various parts of the plant used as natural dyes, including the bark, stems, leaves, roots, seeds, fruits, flowers, and plant sap, to make the colors as in Figure 1. Table 1 shows most of the natural dye sources in Zie Batik are derived from several plant species, including *Rhizophora* spp., *Indigofera tinctoria*, *Maclura cochinchinensis*, *Pelthophorum ferruginum*, *Terminalia belerica*, and *Ceriops condolleana*.

**Table 1.** Dye plant resources for natural batik in the Zie Batik home industry.

| Natural Dyes Plant | Vern Name | Colour | Plant Part | Used Product |
|---|---|---|---|---|
| *Caesalpinia sappan* | Secang | Red | Wood | Dried wood |
| *Ceriops candolleana* | Tingi | Brown | Fruit skin | Boiled * |
| *Maclura cochinchinensis* | Tegeran | Yellow | Bark | Dried bark |
| *Pelthophorum ferruginum* | Soga Jambal | Brown | Bark | Dried bark |
| *Indigofera tinctoria* | Tarum/Indigo | Blue indigo | Leaves, stem | Paste ** |
| *Indigofera arrecta* | Indigo | Blue indigo | Leaves, stem | Paste ** |
| *Rhizopora* spp. | Bakau | Brown | Fruit skin | Boiled * |
| *Strobilantes cusia* | Indigo | Blue indigo | Leaves | Paste ** |
| *Terminalia bellirica* | Jelawe | Yellow | Bark | Dried bark |

Note: The star mark (*) shows application of processing techniques for natural dye used before: * heat processing only; and ** representing microbial fermentation and emulsion processes.

Figure 2 depicts the various organic compounds of the natural dyes with different colors applied to the fabric. The complexity of the coloring technique depends on the design and amount of pigments used to produce the batik.

Zie Batik also continues to develop innovations in natural dye application through their direct (short processing) and indirect usage (long processing). The direct usage was practiced on several dried dye plants, including *C. sappan* wood used to produce a red color, *C. candolleana* fruits in the manufacturing of brown, and the dried bark of *M cochinchinensis* directly produced a yellow. Furthermore, leaves of *I. tinctorial*, *I. arrecta*, and or *S. cusia* were employed after a 3–5-day fermentation and emulsification process to generate an indigo paste; therefore, dissolution in a water ratio of 1:1 (m:v) before usage in the coloring stage is required. The propagule of *Rhizopora* spp. is another material

used that must be boiled for over two hours before utilizing as a brown-light tint. In addition to the single-use, Zie Batik is created by mixing natural dyes to obtain new shades. The combination of these pigments, including *P. ferruginum* and *I. tinctorial*, was employed in the manufacturing of black while *C. sappan* and *C. candolleana* are used for brick-red. This organic pigment mixing relies on basic colors, including red, blue (indigo), and yellow. The addition of these paints produce a dark tone and is used to adjust the brightness level [18].

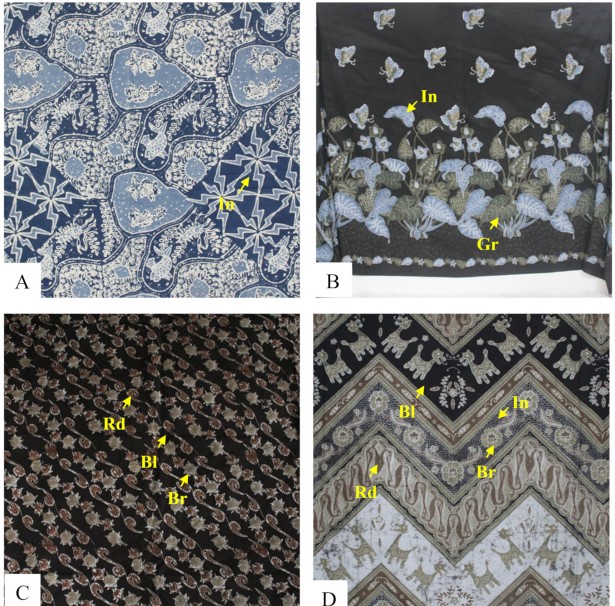

**Figure 2.** Various natural batik of the Zie Batik products by organic material-based natural dyes: (**A**) batik mina (fish motif), one natural dye; (**B**) caladium leaves and butterfly motifs, three natural dyes; (**C**) parang mangrove motif, three natural dyes; and (**D**) Warak Ngendhog motif, four natural dyes. In each batik, the main colors were indigo (In), green (Gr), red (Rd), black (Bl), and brown (Br).

Furthermore, Zie Batik has established a collaboration work with local farmers to build independent crop cultivations because of the natural dye demand to reduce the dependency on coloring plants from other places. This is a system of value chain utilization to reduce production costs from the energy-use sector. The simplification of the business line increases production efficiency and community involvement in actively contributing to the environment's protection [19–21].

*2.2. Clean Production Management Scheme*

The SME developed by Zie Batik is an ideal example of a suitable production implementation during natural dye provision for the batik coloring process. Figure 3 displays the natural dye supply stage performed by the treatment of the required organic materials through drying, boiling, fermentation, and emulsification into ready-to-use coloring materials.

Generally, the waste produced from the process was managed simply, through natural decomposition; also, the timber acquired from the hard wood natural dyer was employed as the fuel, and the resultant ash was used as a fertilizer for the indigo plant (Table 2).

However, the products were not polluted by these methods but by other processes, e.g., the utilization of a mordant followed by fixation produced the bulk of the wastewater, subsequently composed of sodium carbonate ($Na_2CO_3$) and alum ($KAl(SO_4)_2 \cdot 12H_2O$). This challenge is possibly managed through repeated use, while home-scale water waste management systems (WMS) was used for purification, as seen in Figure 4.

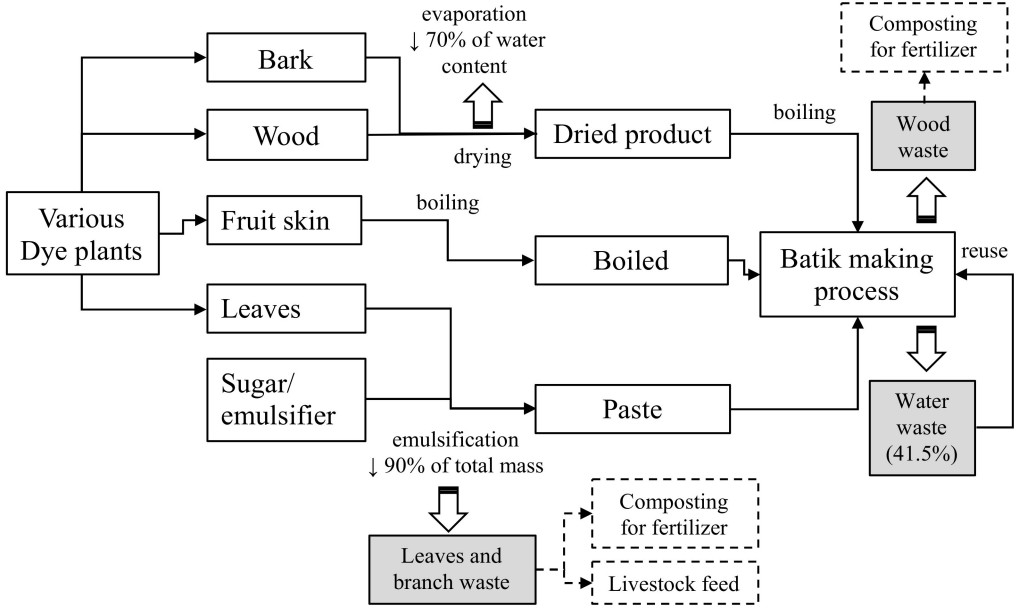

**Figure 3.** Natural dye-producing process and waste management during the coloring stages of the cleaner batik production method implemented by the Zie Batik SME.

**Table 2.** Dye plant sources and waste management.

| Natural Dyes Plant | Sources | Cultivation | By-Product Waste | Waste Management |
|---|---|---|---|---|
| *Caesalpinia sappan* | Semarang, Central Java | Wide distributed but does not cultivated | Wood shavings | Decomposed or firewood |
| *Ceriops candolleana/ C. tagal* | Coastal area of Semarang City | - | No waste | - |
| *Maclura cochinchinensis* | Surakarta, Central Java, originally from Sumatra and Kalimantan | - | Wood pulp | Decomposed |
| *Pelthophorum ferruginum* * | Surakarta, Central Java | - | - | - |
| *Indigofera tinctoria* | - | Local farmers | No waste | - |
| *Indigofera arecta* | - | Local farmers | No waste | - |
| *Rhizopora* spp. | Coastal area of Semarang City | - | No waste | - |
| *Strobilantes cusia* | Japan (was introduced in 2016 for dye plant diversification in Malon) | Local farmers | No waste | - |
| *Terminalia bellirica* | Semarang | - | Wood shavings | Decomposed or timber |

Note: The star mark (*) indicates a rare material that is difficult to get.

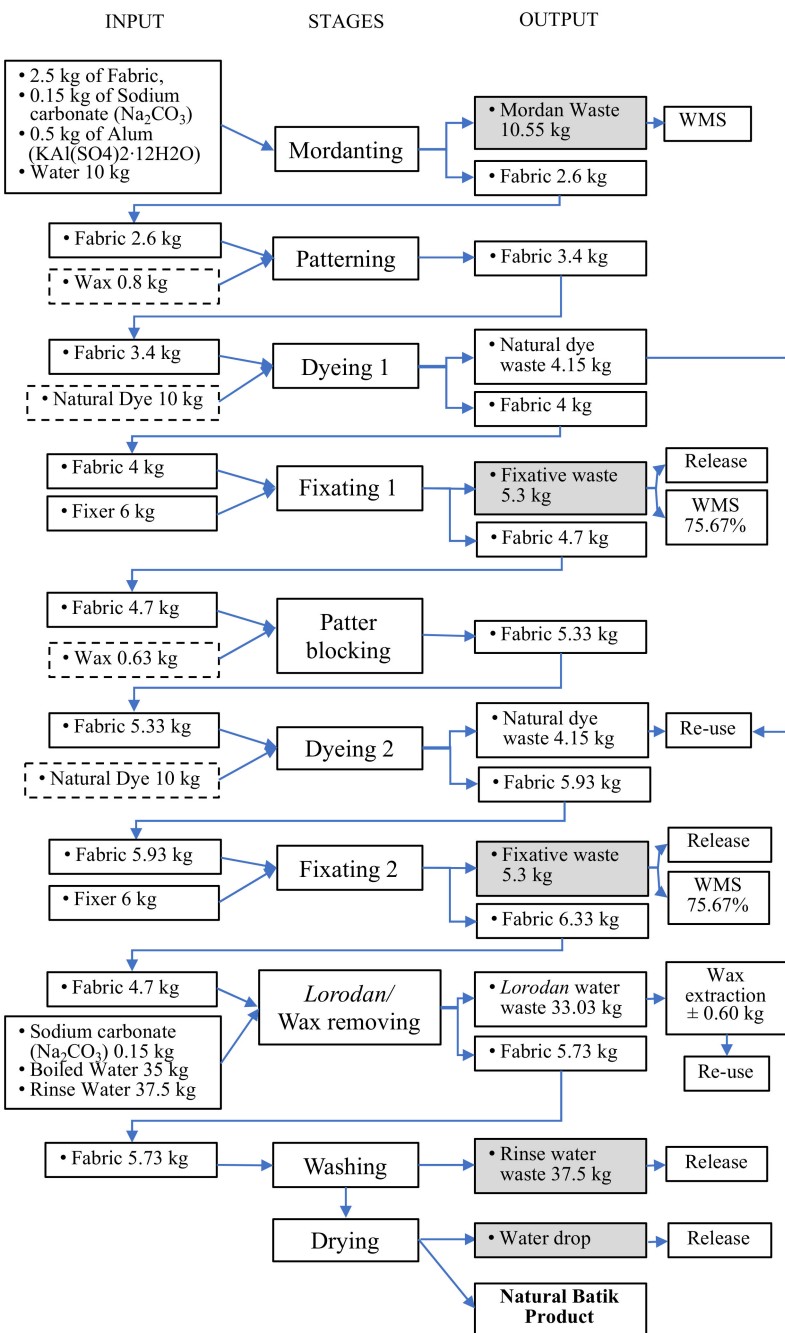

**Figure 4.** Natural dye batik-producing process in Zie Batik. The boxes with dotted lines represent a biodegradable component, and the grey boxes represent a water waste by-product. WMS: water waste management systems.

### 2.3. Clean Production Improvement Strategy

The application of proper maintenance to facilitate clean production is useful as an alternative strategy to surmount inefficiencies. The draining and retention procedures performed at each stage of mordant use, fixation, and desiccation are approaches to reduce environmental pollution and water contamination (Table 3).

**Table 3.** Existing waste management in Zie Batik.

| Step | Problems | Existing Waste Management | Alternative Housekeeping and Waste Management |
|---|---|---|---|
| Mordanting | Wastewater contain $Na_2CO_3$ and $KAl(SO_4)_2 \cdot 12H_2O$ and material of fabric | Reused and deposited in the waste container | Precipitation process to separate solid waste (microfiber) and water purification using a phytoremediation technique |
| Patterning | - | - | - |
| Dyeing 1 and 2 | Natural dye waste | Wood-based dye, the residual product was composted in open areasPaste-based dye was reused until no remaining compound was left | Composting process for wood in a closed and localized place |
| Fixating 1 and 2 | Wastewater contain $CaCO_3$ and $FeSO_4$ | Almost 75% was drained to the environment, and the remaining waste was deposited in the anaerobe-waste container | Streamlining the existing wastewater management installation by increasing the capacity and maintaining the installation |
| Pattern Blocking/waxing | Scattered and evaporated wax | Collected and reused | - |
| *Lorodan*/Wax removing | Wastewater contain wax | Removing process to collect the wax on the surface, and reused for next pattern blocking/waxing | Draining the remaining wastewater to the wastewater management installation |
| Washing and Drying | Water drop on the floor | Discharged into the environment | Applying a water drain collector |

Source: Field observation.

Another problem encountered by Zie Batik in the implementation of clean production was the unreliable production frequency of the natural batik supply. Despite the ease of environmental damage estimation through fixed production agendas, a large magnitude of manufacturing regularity remains dependent on consumer orders. This had the potential of affecting the continuity and availability of the natural batik products in the market. Manufacturing schedules were therefore presented as a solution to preserve the longevity and sustainability of the industry. Furthermore, the execution of planning and appropriate storage and control systems for the dye materials are useful to curb fluctuations and irregular product demands.

## 3. Discussion

The creation of batik with natural dyes is more expensive than with synthetic compounds, as a longer time and advanced skills are required [22]. In addition, the natural dye color is dependent on many factors as a result of the biological origin of the materials. Therefore, batik textiles prepared from organic substances possess different color patterns, bestowing an exclusivity and uniqueness in comparison with artificial materials.

A heightened awareness towards environmental harm has encouraged Zie Batik to implement modifications on products from merely being profit-oriented to performing eco-friendly production activities. The use of organic materials as natural dyes was propelled by concerns of the owner regarding ecological destruction caused by synthetic dye practices. The innovations were executed on the fabric motifs, as well as the types of coloring ingredients used. Initially, organic dyes were exclusively developed from propagules (mangrove fruits) to replace synthetic brown and other dark tones. However, the textile vendor explored dye plants in Semarang and other cities to discover potential sources useful in batik production, although the majority of the necessary organic components are supplied from other districts or provinces in Java Island.

The colorants are essential items in the batik industry and of high quality; neatly executed and attractive motifs executed on comfortable fabrics are expected by clients. [23]. Therefore, the use of eco-friendly organic dye chemicals is currently preferred internationally. These compounds are obtainable from microorganisms including fungi [24], as well as plants, insects, and minerals either directly used or through extraction processes [22]. The natural pigments produced from organic matter are distinctive, as different hues are capable of being generated from a single source [25,26]. However, the use of these starting materials are not devoid of drawbacks and these include difficulty in acquisition due to market unavailability, rapid discoloration, limited color choices [27], minimal stability, pallid appearance, and greater costs in contrast with synthetic dyes [28].

The use of organic materials as an approach to ideal production operations is expected to increase the environmental consciousness of the manufacturer, reduce energy and resource consumption, improve raw material and waste management, and therefore decrease pollution [29]. Moreover, the materials employed in the dry state, including tree barks and woody parts, were dried under sunlight for 5–7 days, and then boiled to extract the concentrated pigments before use. These steps assist the achievement of dye compounds from organic materials, and activate the colors simultaneously, promoting the ease and durability of the dye attachment onto cloth fibers [30,31]. The dehydration of the bark, wood, and fruit is capable of depleting over 70% of the water content, and also increases the length of storage [32]. Meanwhile, during indigo fermentation, an approximately equivalent weight of sucrose was added to generate a denser tint. Sucrose represented the carbon source required for the yeast and bacteria fermentation in the indigo paste preparation process [33]. The organisms involved were fungi from the *Saccharomyces*, *Aspergillus*, *Penicillium*, *Pleurotus*, and *Trametes* genera, [34], as well as several bacteria from the *Alcaligenes*, *Alkalibacterium*, *Amphibacillus*, *Bacillus*, *Corynebacterium*, *Halomonas*, and *Tissierella* genera [33,35,36].

The manufacturing process utilized produced organic waste comprising leaves and stems from the indigo plants. The solid waste had the capacity of being developed into several by-products, including fertilizers and animal feed. Currently, the waste management installation constructed in the Zie Batik industry areas handles solely liquid waste, while the solid remnants are discharged into open areas to facilitate natural decomposition. The composted waste increases the soil nutrient content essential for plant growth [37], and these practices reveal natural dye use as a portion of pollution control requiring zero waste management [38,39].

Mordanting involves the boiling of textiles with chemicals to increase penetration and strengthen the natural dyes' adhesion to the fabric fibers. The use of this solution can sharpens several colors from natural dyes [40–42]. This equally increases the attractiveness of the natural dyes to textile materials and is also useful for producing good color sharpness. Furthermore, swelling increases the affinity of dye, produces a broader color spectrum, and also has better fastness [40]. However, the boiling process in the mordanting stage is useful to remove residual impurities from the weaving process and improve fabric quality. The next stage is fixating, characterized by use of chemical compounds, especially the limestone fixation process ($CaCO_3$) and/or ferrous sulfates ($FeSO_4$). This also has a greater chance of causing environmental pollution, about 75.67% of waste are discharged into the environment and were drained into a management system, as shown in Figure 4.

Furthermore, based on observations, one batch of natural batik in Zie Batik production causes liquid waste worth 95.18 kg per 10 textiles or about 9.518 kg/sheet of batik fabric on average. The amount of pollutants from this industry is higher than the synthetic batik craftsmen's production of water waste in Bogor City [43]. This is probably caused by the inefficient implementation of the clean production process. Meanwhile, based on the concept of clean production, Zie Batik is believed to be inefficient in water usage, while it serves as a main resource in the batik manufacturing process. This outcome is in line with the research of Handayani et al., [44] emphasizing considerable water inefficiency during production of natural dye batik. These colorants are used only once a day, therefore limiting the chances of storage [22]. This is due to the fact that storage of these dyes fades the color and also produces unclear highlighting. In addition, natural dyes are basically organic materials easily degraded either

by temperature, oxidation or bacterial activity [45]. Meanwhile, environment conditions and microbial activities affect the transformation of organic material, and therefore are more environmentally friendly than synthetic dyes.

This study, however, answered the hesitation question of SMEs' ability to develop a core business based on environmental sustainability without ignoring economic and social aspects. However, this hesitation arose because most practitioners assume nature and business aspects conflict when sustainable developments are established. Furthermore, Zie Batik successfully proved to increase the ability of SMEs to act as a driving force in integrating sustainable development aspects through application of clean production. These efforts were made by business actors to improve the quality of this product, and equally attract buyers with creativity, uniqueness, and quality preference. The adaptation eco-friendly practices by this industry promotes more positive employee development and also increase the SMEs' productivity in addition to long-term profitability [46].

Furthermore, the dyeing process of Zie Batik was carried out 3 to 5 times to get the expected color; this is, however, different from synthetic dyes done only 1–2 times [47]. This process equally produces an exothermic reaction, therefore causing the natural dye in the solution to migrate to the fabric to have a balanced concentration [48]; this also increases the concentration of the dye on the fabric when done more often. This technique was used by Zie Batik to produce color gradations and produce more diverse variants in the form of motifs. The rinsing and drying process has not leveled up with the housekeeping standards. Furthermore, some problems, especially inadequate water reservoirs, droplets, and spills, have led to increased liquid waste, in addition to the potential danger caused by slippery floors. The production of this material is not easy to schedule, it being highly dependent on the weather and availability of the organic and fabric materials, in addition to the motif creation. Furthermore, several studies have proved natural batik contains pollutants, especially remaining wax, coloring agents, salts, and fixators; therefore, waste treatment is required before it being discharged into the rivers or drainages [49,50]. This, therefore, outlines the importance of wastewater treatment to decrease environmental pollution [51]

Furthermore, low environmental enforcement and high initial capital costs is the main obstacles towards the implementation of clean production. However, in overcoming these problems, Batik Zie received support from the relevant stakeholders, including local governments, universities, and the private sector, in addition to cooperation built between the SME actors. The campaign of this product through exhibition also continues to serve as promotion by raising the tagline "Batik by natural color".

## 4. Materials and Methods

This investigation was carried out as a case study research [52] with a natural observational explanatory model to describe the clean production process in batik with natural dyes. The data collection method involves a 3-year (2016–2019) observation as part of batik's small and medium-sized enterprises (SMEs) community services programs in Malon Village, Gunungpati, Semarang City, Indonesia. Furthermore, the locus of this study was in Malon Village, with a focus on the Zie Batik SME, due to the following reasons: (1) Zie Batik has been developing clean production since 2006; (2) it was the first SME to develop natural dyes as its core business in Semarang City; (3) it served as the city governments reference for eco-friendly lessons on natural batik; and (4) it provides resource personnel to act as trainers for eco-friendly process learning in natural batik production.

The data was collected by directly observing the dye- and batik-producing process and waste management in Zie Batik's working space. Furthermore, to confirm the data used in this investigation, in-depth interviews were held with 15 respondents—dye plant farmers; 3 respondents—the Zie Batik owners; 5 respondents—the Zie Batik workers; and 15 interim students at Zie Batik. This information was then arranged through a matching process to get the completeness and reliability. The reliable information was however reshaped to match an analyzable data format and was then reduced to eliminate unnecessary data. In addition, the final information was transformed into data and analyzed through categorizing, labeling, and annotating. The final result was interpreted with a narrative

analysis technique. Meanwhile, the data obtained include the raw materials used, mass balance of the batik production process, the produced waste, the obstacles encountered, and the good housekeeping methods applied.

The clean manufacturing process of this industry was formulated through an analysis of the natural dyes' production from organic material. Moreover, the net production of this product includes organic material resource management of the coloring pastes narrowed down normatively. Based on the collected information, this research was, however, focused on the alternative management procedure of water waste in batik production.

## 5. Conclusions

Zie Batik is a small and medium-scale enterprise known to run an eco-friendly batik production business. Furthermore, use of this product serves as an alternative to reduce the impact of environmental pollution caused by synthetic dyes in textile dyeing. This occurs mainly due to the biodegradability, harmlessness, ease of acquisition, and non-toxic liquid waste characteristics of natural dyes. In supporting natural batik production, we found that the color is well-extracted from various plants, especially *C. sappan*, *C. candolleana*, *M. cochinchinensis*, *P. ferruginum*, *I. tinctorial*, *I. arrecta*, *Rhizopora spp.*, *S. cusia*, and *T. bellirica*. Meanwhile, some plant parts, including the bark, stems, leaves, roots, seeds, and sap, are used as natural coloring agents. Furthermore, Zie Batik's net production emphasizes more on the activity of producing dyes and other coloring processes. The remaining natural dyes are used for coloring activities while the organic waste produced was composted naturally and used as timber. Even though the consumption of chemical dyes can be avoided, Zie Batik's use of water in the natural batik-producing process is still wasteful and not well managed. Several improvements should be conducted to minimize the wasting of water and reducing liquid waste discharging into the environment. In addition, regarding Zie Batik's daily production process, they need to enhance the water container volume and build the water treatment plant properly.

In addition, the natural dyes involved as part of the clean production strategy by the Zie Batik SME is capable of manufacturing high-quality, unique, and profitable batik, with a high business value, thus also supporting the social realms beside that of the environmental aspects. However, the use of this product is not left without any disadvantages as it is limited by low color stability, homogeneity, and limited raw material supply compared to synthetic dyes. These limitations affect the continuity of daily production, circumvented by adjusting the manufacturing to meet the consumer's demand.

**Author Contributions:** This research was constructed by a cooperating program. Conceptualization, N.K.T.M. and I.H.; instrument and methodology, I.H., M.M. and R.B.A.; validation, N.K.T.M.; formal analysis, N.K.T.M.; investigation, R.B.A.; resources, N.K.T.M.; data curation, I.H.; writing—original draft preparation, N.K.T.M. and R.B.A.; writing—review and editing, I.H. and M.M.; visualization, R.B.A.; supervision, N.K.T.M.; project administration, M.M.; funding acquisition, N.K.T.M. All authors have read and agreed to the published version of the manuscript.

**Funding:** This study was supported by a project that has received funding from the Deputy of Strengthening Research and Technology, Ministry of Research and Technology of Republic of Indonesia through Community Service Program in the scheme of the Regional Excellence Product Development Program (PPPUD), Research and Community Service Fund (Dana Riset dan Pengabdian Masyarakat—DRPM) for 2018-2020, Grant agreement number 071/SP2H/PPM/DRPM/2020,

**Acknowledgments:** The authors acknowledge Marheno, Zazilah, and Sasi Syifaurohmi as the owners of Zie Batik who have provided permission, assistance, and a place for the authors to conduct this research, and then allowed to publish the paper.

**Conflicts of Interest:** The authors declare no conflict of interest. The funders had no role in the design of the study; in the collection, analyses, or interpretation of data; in the writing of the manuscript, or in the decision to publish the results.

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
