# Peer review of "Organic Material for Clean Production in the Batik Industry: A Case Study of Natural Batik Semarang, Indonesia"

_recycling, doi:10.3390/recycling5040028_

Round 1
Reviewer 1 Report
Manuscript number: Recycling 883997
Manuscript title: Organic Material as Clean Production in Batik Industry: A Case Study of Natural Batik Semarang, Indonesia
The authors present a case study for the application of organic materials as natural dyes in clean production of textile industry to sustain the environment.
- In my opinion, it is less a research than a description of a present activity. The authors must clearly describe the objectives, novelty and originality of their research.
- The presented information is minor, more from literature than from the authors’ research, e.g. “During the process, color particles migrate from the solution into the fabric and absorbed by the fabric fibers.”. The same observation even for “Conclusions”, e.g. “Plant parts that can be used as natural coloring agents include bark, stems, leaves, roots, seeds, and sap.”.
- The grammatical errors must be corrected.
- The wrong chemical formulas must be corrected, e.g. “Ca2OH”, “CaCO2”.
Author Response
Thank you for your constructed review, that spotting information missing in my manuscript title: Organic Material as Clean Production in Batik Industry: A Case Study of Natural Batik Semarang, Indonesia.
I would like to reply for some review includes:
- In my opinion, it is less a research than a description of a present activity. The authors must clearly describe the objectives, novelty and originality of their research.
reply: we have already added some information to emphasis the the objectives, novelty and originality of their research in the introduction line 58-68 highlighted by yellow. the blue highlight is answers for 2nd reviewer
- The presented information is minor, more from literature than from the authors’ research, e.g. “During the process, color particles migrate from the solution into the fabric and absorbed by the fabric fibers.”. The same observation even for “Conclusions”, e.g. “Plant parts that can be used as natural coloring agents include bark, stems, leaves, roots, seeds, and sap.”.
reply: the synthetic information in line 279 using reference information because the Zie batik ony using natural dye and never experienced with synthetic or chemical dye. the we delete "During the process, color particles migrate from the solution into the fabric and absorbed by the fabric fibers" to give more precious information to explain how the coloring process. It just to explain the batik-producing process.
- The grammatical errors must be corrected.
reply: has been improved
- The wrong chemical formulas must be corrected, e.g. “Ca2OH”, “CaCO2”
reply: has been revised
please see the attachment bellow.
thank you for your enlightenment.

Reviewer 2 Report
The manuscript "Organic Material as Clean Production in Batik Industry: A Case Study of Natural Batik Semarang, Indonesia" describes the use of organic materials as natural dyes in clean production of textile industry to protect the environment. The goals and motivation of the study are quite clear. The rewritten manuscript can be accepted for publication after considering the issues outlined below.
Additional comments/points that need to be addressed:
- Line 28 and 29 and 132: The formula of compounds in the text is wrong for example:
“of the wastewater containing sodium carbonate (Na2CO3), alum (KAl(SO4)2.12H2O), and fixatives (Ca2OH and FeSO4) that were discharged into environment but needed more processing treatment.”
should be: “of the wastewater containing sodium carbonate (Na2CO3), alum (KAl(SO4)2 ·12H2O), and fixatives (Ca(OH)2 and FeSO4) that were discharged into environment but needed more processing treatment.”
- Line 132: (KAl(SO4)2.12H2O) shoud be “(KAl(SO4)2 12H2O)”
- What kind of compound is CaCO2? Maybe, the authors meant CaCO3.
- The subchapter 2.1 Natural Batik Profile does not presents any results and should be included in the introduction. The audience can have similar impression reading the chapter 3.
- Two tables have the same numbers. Line 127 and 147.
Author Response
Thank you for your constructive review, that spotting the information missing in my manuscript title: Organic Material as Clean Production in Batik Industry: A Case Study of Natural Batik Semarang, Indonesia.
The goals and motivation of the study are quite clear. The rewritten manuscript can be accepted for publication after considering the issues outlined below.
reply : thank you
Additional comments/points that need to be addressed:
- Line 28 and 29 and 132: The formula of compounds in the text is wrong for example:
“of the wastewater containing sodium carbonate (Na2CO3), alum (KAl(SO4)2.12H2O), and fixatives (Ca2OH and FeSO4) that were discharged into environment but needed more processing treatment.”
should be: “of the wastewater containing sodium carbonate (Na2CO3), alum (KAl(SO4)2 ·12H2O), and fixatives (Ca(OH)2 and FeSO4) that were discharged into environment but needed more processing treatment.”
- Line 132: (KAl(SO4)2.12H2O)
- shoud be “(KAl(SO4)212H2O)”
reply : has been revised
- What kind of compound is CaCO2? Maybe, the authors meant CaCO3.
reply : has been revised
- The subchapter 2.1 Natural Batik Profile does not presents any results and should be included in the introduction. The audience can have similar impression reading the chapter 3.
reply : has been moved and revised
- Two tables have the same numbers. Line 127 and 147.
reply : has been revised
please see the attachment bellow.
thank you
